# Universal Aspect of Relaxation Dynamics in Random Spin Models

Tian-Gang Zhou,[1] Wei Zheng,[2, 3, 4] and Pengfei Zhang[5, *]

[1]*Institute for Advanced Study, Tsinghua University, Beijing, 100084, China*
[2]*Hefei National Research Center for Physical Sciences at the Microscale and School of Physical Sciences,*
*University of Science and Technology of China, Hefei 230026, China*
[3]*CAS Center for Excellence in Quantum Information and Quantum Physics,*
*University of Science and Technology of China, Hefei 230026, China*
[4]*Hefei National Laboratory, University of Science and Technology of China, Hefei 230088, China*
[5]*Department of Physics, Fudan University, Shanghai, 200438, China*
(Dated: May 5, 2023)

The concept of universality is a powerful tool in modern physics, allowing us to capture the essential features of a system's behavior using a small set of parameters. In this letter, we unveil universal spin relaxation dynamics in anisotropic random Heisenberg models with infinite-range interactions at high temperatures. Starting from a polarized state, the total magnetization can relax monotonically or decay with long-lived oscillations, determined by the sign of a universal single function $A = -\xi_1^2 + \xi_2^2 - 4\xi_2\xi_3 + \xi_3^2$. Here $(\xi_1, \xi_3, \xi_3)$ characterizes the anisotropy of the Heisenberg interaction. Furthermore, the oscillation shows up only for $A > 0$, with frequency $\Omega \propto \sqrt{A}$. To validate our theory, we compare it to numerical simulations by solving the Kadanoff-Baym (KB) equation with a melon diagram approximation and the exact diagonalization (ED). The results show our theoretical prediction works in both cases, regardless of a small system size $N = 8$ in ED simulations. Our study sheds light on the universal aspect of quantum many-body dynamics.

*Introduction.–* A complete description of realistic many-body systems always contains a large number of parameters. For example, typical solid-state material contains complicated interactions between electrons, phonons, nuclei, and impurities. However, properties that are of physical interest can usually be captured by simple toy models with few parameters. This is a remarkable consequence of universality. The universality states that microscopically different systems can share the same physics at large scales. It usually emerges in the low-energy limit. For example, the phase transition of many-body systems can be classified into universality classes determined only by the symmetry and dimension of systems [1, 2]. Low-energy scattering between atoms can be well described by a single parameter, the scattering length $a_s$, despite details of underlying microscopic interaction potentials [3]. Aiming at deepening our understanding of realistic systems, discovering new universalities becomes an important subject in modern quantum many-body physics.

Recent years have witnessed a great breakthrough in understanding novel quantum dynamics in many-body systems both theoretically [4–30] and experimentally [31–37]. In the previous studies on relaxation, most universal dynamics emerges in the low temperature or long time regime. That reflects the microscopic details of models are smoothed out in the low energy scale. However at high temperature and short time, it is common believed that most microscopic details are involved in the evolution. Such that the evolution is highly model dependent and hard to observe a universal dynamics. In this work, we unveil that a universal aspect of relaxation dynamics which shows up in an anisotropic Heisenberg model with all-to-all interactions even at high temperatures and short time. The Hamiltonian reads:

$$\hat{H} = \sum_{1 \le i < j \le N} J_{ij}(\xi_1 \hat{S}_i^x \hat{S}_j^x + \xi_2 \hat{S}_i^y \hat{S}_j^y + \xi_3 \hat{S}_i^z \hat{S}_j^z) - h(t) \sum_{1 \le i \le N} \hat{S}_i^x. \quad (1)$$

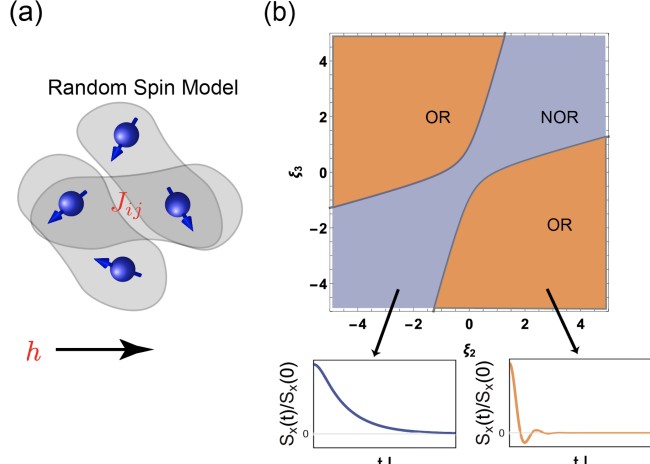

FIG. 1. (a). Schematics of the random spin model with random (anisotropic) Heisenberg interactions $J_{ij}$ in the magnetic field $h$. (b). Different dynamical behaviors of the system for different anisotropy parameters $(1, \xi_2, \xi_3)$. The boundary line is determined by $A = -\xi_1^2 + \xi_2^2 - 4\xi_2\xi_3 + \xi_3^2 = 0$, which is symmetric under the reflection along $\xi_2 = \pm\xi_3$. OR and NOR denote the oscillating regime and non-oscillating relaxation regime respectively, distinguished by features of the magnetization relaxation process.

This model with different anisotropy parameters $(\xi_1, \xi_2, \xi_3)$ has been realized in cold molecules [38, 39], NV centers [40, 41], trapped fermions [42], Rydberg atoms [43, 44], high spin atoms [45], and solid-state NMR systems [46, 47]. A schematic figure is presented in FIG. 1 (a). Because of random locations or complicated spatial wavefunctions of spin carriers, $J_{ij}$ is usually modeled as independent random Gaussian variables with expectation $\overline{J_{ij}} = \bar{J}/N$ and variance $\overline{\delta J_{ij}^2} = 4J^2/N$.

We focus on the following protocol: The system is prepared at high temperatures with a polarization field $h(t < 0) = h$, which induces a magnetization in the $x$ direction. We then monitor the relaxation of the total magnetization after turning off $h$ suddenly at $t = 0$. We find the total magnetization decays either monotonically or with long-lived oscillations, depending on $A = -\xi_1^2 + \xi_2^2 - 4\xi_2\xi_3 + \xi_3^2$. The oscillation only appears for $A > 0$, in which case the frequency satisfies $\Omega \propto J\sqrt{A}$. Importantly, this phenomenon should be understood as a universal property of the relaxation dynamics since the criterion only contains a specific combination of anisotropic parameters, instead of full details of the microscopic model (1). To validate our theoretical prediction, we further perform numerical simulations based on the Kadanoff-Baym (KB) equation with melon-diagram approximations, and the exact diagonalization (ED). Numerical results show the theoretical prediction works in both cases, although we are limited to a small system size $N = 8$ in the ED. Our work also provides a novel theoretical framework to analyze the dynamics of randomly interacting quantum spin models.

*Theoretical Analysis.–* We are interested in the relaxation dynamics of total magnetization. Our theoretical analysis is based on the path-integral approach on the Keldysh contour, as elaborated in [48, 49]. To begin with, we observe that the random spin model can be written in terms of Abrikosov fermion operators $\hat{c}_{i,s}$ with spins $s = \uparrow, \downarrow$ in the single occupation subspace. Explicitly, we have $\hat{S}_i^\alpha = \sum_{ss'} \frac{1}{2}\hat{c}_{i,s}^\dagger (\sigma^\alpha)_{ss'}\hat{c}_{i,s'}$, where $\alpha = x, y, z$ and $\sigma^\alpha$ denote the corresponding Pauli matrices. Since the Hamiltonian (1) exhibits $\pi$ rotation symmetries along the $x$ axis, the total magnetization can only be along the $x$ axis. We thus introduce $m(t) \equiv \langle \hat{S}^x(t) \rangle$. After imposing the symmetry constraints [50], the magnetization can be computed by real-time Green's functions of fermion operators:

$$m(t) = -iG_{\uparrow\downarrow}^>(t, t) = -iG_{\uparrow\downarrow}^<(t, t), \tag{2}$$

where we have defined $G_{ss'}^>(t_1, t_2) \equiv -i \sum_l \left\langle c_{l,s}(t_1)c_{l,s'}^\dagger(t_2) \right\rangle / N$ and $G_{ss'}^<(t_1, t_2) \equiv i \sum_l \left\langle c_{l,s'}^\dagger(t_2)c_{l,s}(t_1) \right\rangle / N$.

The relaxation dynamics of $m(t)$ can then be computed once we obtain the Green's functions $G^{\gtrless}(t_1, t_2)$. It is known that the evolution of $G^{\gtrless}(t_1, t_2)$ is governed by the Kadanoff-Baym equation, which can be derived by the Schwinger-Dyson equation on the Schwinger-Keldysh contour.

$$i\partial_{t_1}G^{\gtrless} + \frac{1}{2}h_{\text{eff}}(t_1)\sigma^x G^{\gtrless} = \Sigma^R \circ G^{\gtrless} + \Sigma^{\gtrless} \circ G^A,$$
$$-i\partial_{t_2}G^{\gtrless} + \frac{1}{2}h_{\text{eff}}(t_2)G^{\gtrless}\sigma^x = G^R \circ \Sigma^{\gtrless} + G^{\gtrless} \circ \Sigma^A. \tag{3}$$

Here we have introduced self-energies $\Sigma^{\gtrless}$ and $\Sigma^{R/A}$. We define the operation $\circ$ for functions with two time variables as $f \circ g \equiv \int dt_3\, f(t_1, t_3)g(t_3, t_2)$. The retarded and advanced Green's functions $G^{R/A}$ are related to $G^{\gtrless}$ by $G^{R/A} = \pm\Theta(\pm t_{12})(G^> - G^<)$, where $\Theta(t)$ is the Heaviside step function. Similar relations work for self-energies $\Sigma^{R/A}$. $h_{\text{eff}}(t) = h(t) + \bar{J}m(t)$ is the effective magnetic field, which includes

the mean-field contribution from $\bar{J}$. For $t < 0$, the system is prepared in thermal equilibrium. Consequently, we have $G^{\gtrless}(t_1, t_2) = G_\beta^{\gtrless}(t_{12})$ for $t_1, t_2 < 0$. For either $t_1 > 0$ or $t_2 > 0$, the Green's functions evolve due to the quantum quench and should be obtained by solving Eq. (3) after the self-energy is specified.

The approximation comes in when we try to relate the self-energies to Green's functions. After transforming into the Abrikosov fermion representation, the random Heisenberg interaction takes the form of random fermion scatterings. Interestingly, such random interaction terms is a close analog of the celebrated complex Sachdev-Ye-Kitaev (SYK) model [51–57]. Motivated by this observation, here we make the melon diagram approximation for the fermion self-energy. A formal argument to control errors is to generalize the Hamiltonian (1) into large-$M$ spins, as in the seminal work by Sachdev and Ye [58]. Consequently, we have

$$\Sigma^{\gtrless}(t_1, t_2) = \frac{J^2}{4} \sum_{\alpha,\alpha'} \xi_\alpha\xi_{\alpha'}$$
$$\sigma^{\alpha'}G^{\gtrless}(t_1, t_2)\sigma^\alpha \text{Tr}\left[\sigma^{\alpha'}G^{\gtrless}(t_1, t_2)\sigma^\alpha G^{\lessgtr}(t_2, t_1)\right], \tag{4}$$

Here $\alpha, \alpha' \in \{1, 2, 3\}$ and we omit spin indices for conciseness. We have introduced the anisotropy vector $\boldsymbol{\xi} = (\xi_1, \xi_2, \xi_3)$. The melon diagram approximation may fail in the low-temperature limit if the system exhibits spin glass orders [59]. In this work, we avoid this problem by focusing on the high-temperature regime with $\beta J \ll 1$. Combining Eq. (2), (3), and (4) leads to a set of closed equations which determines the relaxation of the magnetization.

Typical numerical results for $m(t)$ obtained by two methods are shown in FIG. 2. Here we consider examples with $\xi_1 = 1$ and $(\xi_2, \xi_3) = (0.8, -1.5), (0.8, 1.5), (1, -2)$, and $(1, 2)$. We set the initial temperature $\beta J = 0.04$, the polarization field $h/J = 10$ and $\bar{J} = 0$. In the long-time limit, the system exhibits the quantum thermalization to the thermal ensemble with $h = 0$. In this case, $\pi$ rotations along $y$ or $z$ also become the symmetry of the Hamiltonian, which makes $m(\infty) = 0$. According to the relaxation process, different anisotropy parameters can be divided into two groups, under which $m(t)$ relaxes monotonically (for $(\xi_2, \xi_3) = (0.8, 1.5)$ and $(1, 2)$) or with long-lived oscillations (for $(\xi_2, \xi_3) = (0.8, -1.5)$ and $(1, -2)$). Furthermore, we numerically checked that the presence of the oscillation is stable against deformations of parameters. As a result, we propose the Hamiltonian (1) with different $(\xi_1, \xi_2, \xi_3)$ can be separated in parameter regimes with oscillating relaxation (OR) versus non-oscillating relaxation (NOR), as shown in FIG. 1. In supplementary material [50], we verify that the difference in the dynamical behavior can not be detected in equilibrium, for example, via spin susceptibility.

*Oscillation mode.–* After evolving for a long time, the total magnetization, as well as off-diagonal components of Green's functions, becomes very small. Consequently, we can perform a linearized analysis of the KB equation to reveal the mechanism for the oscillation and determine the criterion for differ-

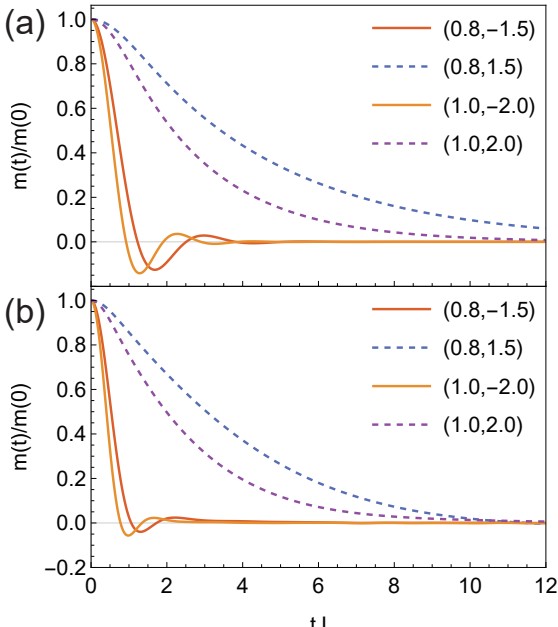

FIG. 2. The numerical result for the evolution of the magnetization $m(t)$ by numerically solving: (a) the Kadanoff-Baym equation (3) and (b) exact diagonalization with system size $N = 8$. Initially, the system is in thermal equilibrium with $\beta J = 0.04$, $\bar{J} = 0$ and $h/J = 10$. We take $\xi_1 = 1$ and consider four different anisotropy parameters $(\xi_2, \xi_3) = (0.8, -1.5)$, $(0.8, 1.5)$, $(1, -2)$, and $(1, 2)$, which corresponds to $A = 6.69$, $-2.91$, $12$, and $-4$. The results show that the relaxation of $m(t)$ is monotonic/oscillating if $A < 0/A > 0$. These two numerical results match each other to good precision despite a small $N$.

ent dynamical behaviors. A differential equation that governs the long-time evolution of the magnetization can be derived following a few steps:

**Step 1.–** The linearized analysis can be largely simplified after the Keldysh rotation. We introduce the standard Keldysh Green's function of fermions as $G^K = G^> + G^<$. The total magnetization can be expressed as its off-diagonal component:

$$m(t) = -iG^K_{\uparrow\downarrow}(t,t)/2. \tag{5}$$

We can further combine equations in (3) to derive the equation for $G^K$. The result reads

$$G^K = G^R \circ \Sigma^K \circ G^A, \quad \text{with } \Sigma^K = \Sigma^> + \Sigma^<. \tag{6}$$

**Step 2.–** We linearize Eq. (6) around the equilibrium solution in the long-time limit after the quantum thermalization. We expand $G^a(t_1, t_2) = G^{a,\beta_f}(t_{12}) + \delta G^a(t_1, t_2)$, where $G^{a,\beta_f}(t)$ is the equilibrium Green's function on the final state. Leaving details into the supplementary material [50], the off-diagonal element of (6) reads

$$\delta G^K_{\uparrow\downarrow} = G^{R,\beta_f}_{\uparrow\uparrow} \circ \delta\Sigma^K_{\uparrow\downarrow} \circ G^{A,\beta_f}_{\uparrow\uparrow},$$
$$\delta\Sigma^K_{\uparrow\downarrow} = \frac{1}{4}J^2 A\left((G^{>,\beta_f}_{\uparrow\uparrow})^2 + (G^{<,\beta_f}_{\uparrow\uparrow})^2\right)\delta G^K_{\uparrow\downarrow}. \tag{7}$$

where we have introduced $A = -\xi_1^2 + \xi_2^2 - 4\xi_2\xi_3 + \xi_3^2$ as in the introduction. Since $G^{K,\beta_f}_{\uparrow\downarrow} = 0$, Eq. (5) is equivalent to $m(t) = -i\delta G^K_{\uparrow\downarrow}(t,t)/2$.

**Step 3.–** To proceed, we need to obtain approximations for $G^{a,\beta_f}_{\uparrow\uparrow}$. In thermal equilibrium with $h = 0$, the self-energies (4) can be simplified as

$$\Sigma^{\gtrless,\beta_f}_{ss'}(t) = -\frac{J^2\xi^2}{2}G^{\gtrless,\beta_f}_{ss}(t)^3\delta_{ss'}, \tag{8}$$

where we have used $G^{>,\beta_f}_{ss}(t) = -G^{<,\beta_f}_{ss}(-t)$ due to the particle-hole symmetry. Eq. (8) then matches the self-energy of the Majorana SYK$_4$ model with effective coupling constant $J|\xi|/\sqrt{2}$. It is known that at high temperatures $\beta J \ll 1$, the SYK model can be described by weakly interacting quasi-particles [60]. Taking the Lorentzian approximation, we have

$$G^{R/A,\beta_f}_{\uparrow\uparrow}(t) \approx \mp i\Theta(\pm t)e^{-\Gamma|t|/2}, \quad G^{\gtrless,\beta_f}_{\uparrow\uparrow}(t) \approx \mp ie^{-\Gamma|t|/2}/2. \tag{9}$$

with quasi-particle decay rate $\Gamma \propto J$.

**Step 4.–** Finally, combining (7) and (9), $\delta G^K_{\uparrow\downarrow}$ satisfies the differential equation

$$\left(\partial_{t_1} + \frac{\Gamma}{2}\right)\left(\partial_{t_2} + \frac{\Gamma}{2}\right)\delta G^K_{\uparrow\downarrow} = -\frac{A}{8}J^2 e^{-\Gamma|t_{12}|}\delta G^K_{\uparrow\downarrow}. \tag{10}$$

We have inversed the retarded/advanced Green's functions (9) using $\left(\partial_t + \frac{\Gamma}{2}\right)G^{R,\beta_f}_{\uparrow\uparrow}(t) = \left(\partial_t - \frac{\Gamma}{2}\right)G^{A,\beta_f}_{\uparrow\uparrow}(t) = -i\delta(t)$.

Eq. (10) is the starting point for analyzing the relaxation dynamics. Since it is invariant under time translations, we separate out the center-of-mass time dependence by introducing $\delta G^K_{\uparrow\downarrow}(t_1, t_2) = \text{Re } e^{-\lambda\frac{t_1+t_2}{2}}\varphi(t_{12})$. The relaxation is oscillatory only if $\lambda$ is complex. Interestingly, $\varphi(t_{12})$ then satisfies the 1D Schrödinger equation

$$-\frac{(\Gamma-\lambda)^2}{4}\varphi(t_{12}) = -\partial^2_{t_{12}}\varphi(t) + \frac{A}{8}J^2 e^{-\Gamma|t_{12}|}\varphi(t_{12}), \tag{11}$$

where $-\frac{(\Gamma-\lambda)^2}{4}$ plays the role of the energy $E$ and $\frac{A}{8}J^2 e^{-\Gamma|t_{12}|}$ plays the role of potential $V$. Eq. (11) suggests the boundary line between the oscillating regime and the non-oscillating regime is at $A = 0$: For $A < 0$, the potential energy is negative. It is known that in 1D any attractive potential exhibits at least one bound state. Denoting the energy of the ground state as $-|E_0|$, we can solve $\lambda = \Gamma - 2\sqrt{|E_0|}$, which is real. Consequently, we expect the magnetization relaxes monotonically. For $A > 0$, the potential is repulsive. The eigenstates of the (11) are scattering modes with continuous positive energy $E$. We find $\lambda = \Gamma \pm 2i\sqrt{E}$, which is complex. This leads to oscillations in the relaxation process.

To further determine the typical oscillation frequency $\Omega$, we need to determine the typical energy $E$ that contributes to the quench dynamics. According to Eq. (5), the magnetization probes the decay of the wave function at $t_{12} = 0$, where the potential energy is $\sim AJ^2$. For $E \ll AJ^2$, the eigenstate has exponentially small weight near $t_{12} = 0$. As a result, the

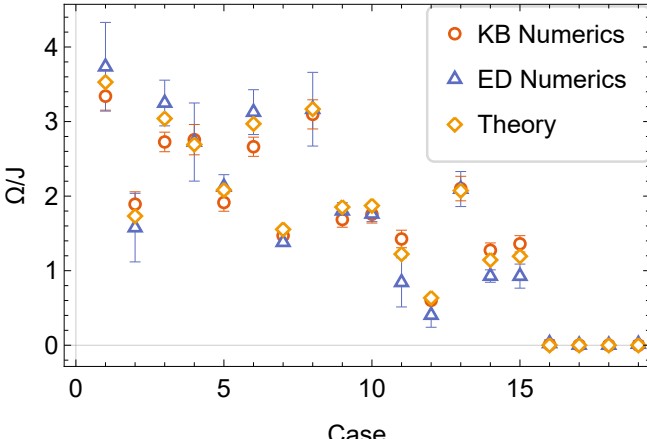

FIG. 3. A comparison between the theoretical prediction $\Omega_{Th}/J = c_0 \sqrt{A}$ and numerical simulations. Here we choose $c_0 = 1$. In each case, we randomly choose the anisotropic parameter $(\xi_1, \xi_2, \xi_3)$. Initially, the system is also in thermal equilibrium with $\beta J = 0.04$, $\bar{J} = 0$, and $h/J = 10$. The numerical data are obtained by fitting numerics based on the KB equation and ED method. The error bars correspond to standard deviations when concerning the different fitting regions.

corresponding contribution to $m(t)$ can be neglected. We can approximate

$$m(t) \sim \int_{AJ^2} dE\, c(E) e^{-\Gamma t - 2i \sqrt{E} t}. \tag{12}$$

Here $c(E)$ is some smooth function determined by the initial condition. We then expect $\Omega \approx c_0 \sqrt{A} J$, with some $O(1)$ constant $c_0$ which does not depend on parameters in the Hamiltonian (1) and should be extracted using numerics. Interestingly, the result predicts the oscillation period $T = 2\pi/\Omega$ diverges as we approach $A = 0$, which can be viewed as an analog of the divergence of the correlation length in traditional phase transition described by order parameters.

We comment that our results unveil the universality of relaxation dynamics in random spin models. Although the microscopic model in (1) contains several parameters, the criterion for the different relaxation behaviors, as well as the oscillation frequency, only depends on a specific combination $A$. This is a direct analog of universality in the scattering theory, where for a complicated potential, the low-energy scattering problem can only depend on a specific combination of microscopic parameters, which is the scattering length.

*Numeric tests.–* Now we compare the theoretical predictions in the last section to numerics. We can firstly test our criteria using numerics by solving the KB equation. After fixing $\xi_1 = 1$, we find $A = 6.69, -2.91, 12$, and $-4$ respectively for $(\xi_2, \xi_3) = (0.8, -1.5), (0.8, 1.5), (1, -2)$, and $(1, 2)$. Our theory then predict oscillating relaxation for $(0.8, -1.5), (1, -2)$ and non-oscillating relaxation for $(0.8, 1.5), (1, 2)$. This is consistent with the evolution of $m(t)$ as shown in FIG. 2(a). To test our theoretical prediction beyond the melon diagram approximation, we also perform numerical simulations using the

ED for a system size $N = 8$ and 1000 random realizations. The system was initially prepared in thermal equilibrium with $\beta J = 0.04$ and $h/J = 10$. We choose anisotropy $\xi_1 = 1$, $(\xi_2, \xi_3) = (0.8, -1.5), (0.8, 1.5), (1, -2)$, and $(1, 2)$, as in the previous section. Despite using a small $N$, FIG. 2(b) shows the oscillating/non-oscillating relaxation in the ED matches the prediction of the KB equation.

We further compare our prediction of the oscillation frequency $\Omega \approx c_0 \sqrt{A} J$ to numerical results. We obtain $\Omega$ in numerics by fitting $m(t) = m_0 \cos(\Omega t + \theta) e^{-\Gamma t} + m_{\text{offset}}$. Here $m_0$ is the amplitude, $\theta$ is the phase, $\Gamma$ is the quasi-particle decay rate, and $m_{\text{offset}}$ is the offset which is significant in the finite $N$ ED numerics. The fitting particularly focuses on the matching in the small $m(t)$ region. Hence the detailed fitting region and the error bars caused by such ambiguity are left to the supplementary material [50]. The results are shown in FIG. 3. We randomly choose the anisotropic parameters, and the first 15 cases correspond to $A > 0$, and the last 4 cases to $A < 0$[50]. Among the $A > 0$ cases, the mean ratio between the numerical data and the polynomial $A$ reads $\overline{\Omega_{\text{KB}}/(J \sqrt{A})} = 0.995 \pm 0.018$ and $\overline{\Omega_{\text{ED}}/(J \sqrt{A})} = 0.94 \pm 0.04$. Therefore, we set $c_0 = 1$ for theoretical predictions in FIG 3. Although the error bars for ED numerics are significantly larger than KB numerics since the calculation is based on the finite $N = 8$ system, we find the theoretical prediction of the oscillation frequency almost matches the KB results and the ED results, up to the error bars. From FIG. 3, most notably, the OR and NOR relaxation are sharply distinguished by the $A > 0$ or $A < 0$ criterion, which is perfectly aligned with our theoretical analysis.

*Discussions.–* In this work, we show that the random Heisenberg model with all-to-all interactions exhibits universal relaxation dynamics governed by a single parameter $A = -\xi_1^2 + \xi_2^2 - 4\xi_2\xi_3 + \xi_3^2$. Unlike traditional examples where the universality emerges in the low-energy limit, here the universal physics appears at high temperatures. For $A < 0$, the magnetization decays monotonically after we turn off the polarization field. For $A > 0$, long-lived oscillation appears during the relaxation process, with a frequency $\Omega \propto J \sqrt{A}$. Our theoretical analysis is based on the path-integral approach on the Keldysh contour, which is verified by comparing our theory to numerical simulations by solving the KB equation or the ED.

We remark that quantum coherence is essential for the existence of the oscillating relaxation regime. As an example, if we spoil the coherence by considering time-dependent random interactions instead of static interactions, the magnetization is expected to decay monotonically: After replacing $J_{ij}$ with Brownian variables $J_{ij}(t)$, Eq. (10) is replaced by

$$\left(\partial_t + \Gamma\right) \delta G_{\uparrow\downarrow}^K(t, t) = -\frac{AJ}{8} \delta G_{\uparrow\downarrow}^K(t, t), \tag{13}$$

as derived in the supplementary material [50]. This results in $m(t) \sim e^{-(\Gamma + AJ/8)t}$ with a simple exponential decay, on contrary to the existence of different dynamical behaviors in the static case.

We also point out that amazingly our criteria $A > 0$ for the oscillation regime matches the criteria proposed in [57] for the presence of the instability towards the formation of wormholes with $\xi_1 = \xi_2 = 1$. However, the analysis in [57] focuses on the low-temperature regime, while in this work we focus on high temperatures. This makes it difficult to establish a direct relationship between the two theoretical analyses. It would be interesting if there is some version of duality between the high-temperature and the low-temperature limit. Since the wormhole phase is non-chaotic, it is also interesting to study the out-of-time correlator or the operator size distribution in regimes with different dynamical behaviors.

On the experimental side, the quench experiment recently performed in NMR systems directly measured the magnetization evolution when turning off the external field, and our results can be straightforwardly verified in this experiment[61].

*Note Added.* Universal behaviors of auto-correlation function related to the quench dynamics discussed here, including oscillatory versus non-oscillatory behavior, have been related to Lanczos coefficients computed for determining the Krylov complexity in Ref. [62].

*Acknowledgment.* We are especially grateful to the invaluable discussions with Hui Zhai, whose advice is indispensable for the whole work. We thank Riqiang Fu, Yuchen Li, Xinhua Peng, Xiao-Liang Qi and Ren Zhang for their helpful discussions. PZ is partly supported by the Walter Burke Institute for Theoretical Physics at Caltech.

––––––––––––

* PengfeiZhang.physics@gmail.com

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

# Supplementary Material: Universal Aspect of Relaxation Dynamics in Random Spin Models

Tian-Gang Zhou,[1] Wei Zheng,[2, 3] and Pengfei Zhang[4, *]

[1] *Institute for Advanced Study, Tsinghua University, Beijing, 100084, China*
[2] *Hefei National Laboratory for Physical Sciences at the Microscale and Department of Modern Physics,*
*University of Science and Technology of China, Hefei 230026, China*
[3] *CAS Center for Excellence in Quantum Information and Quantum Physics,*
*University of Science and Technology of China, Hefei 230026, China*
[4] *Department of Physics, Fudan University, Shanghai, 200438, China*
(Dated: May 2, 2023)

In this Supplementary Material, we present: (1) The formulation of the equilibrium Green's function; (2) The symmetry of the Green's function; (3) Details of the numerical simulation; (4) Explicit calculation of the self energy; (5) Derivation of the normal mode equation with regular interactions; (6) Derivation of the normal mode equation with Brownian interactions; (7) Susceptibility in thermal equilibrium.

## I. THE FORMULATION OF THE EQUILIBRIUM GREEN'S FUNCTION

To further proceed, we need the real-time approach. We begin with the greater and lesser Green's functions in real-time defined as

$$
\begin{aligned}
G_{ss'}^{>}(t_1, t_2) &\equiv -i\frac{1}{N}\sum_l \left\langle c_{l,s}(t_1)c_{l,s'}^{\dagger}(t_2) \right\rangle, \\
G_{ss'}^{<}(t_1, t_2) &\equiv i\frac{1}{N}\sum_l \left\langle c_{l,s'}^{\dagger}(t_2)c_{l,s}(t_1) \right\rangle,
\end{aligned}
\tag{1}
$$

as well as the retarded and advanced Green's function defined as

$$
G_{ss'}^{R/A}(t_1, t_2) \equiv \mp i\Theta\left(\pm t_{12}\right)\frac{1}{N}\sum_l \left\langle \{c_{l,s}(t_1), c_{l,s'}^{\dagger}(t_2)\} \right\rangle,
\tag{2}
$$

where $\Theta\left(t\right)$ is the Heaviside step function and we have defined $t_{12} = t_1 - t_2$. In the thermal equilibrium, all Green's functions are only functions of $t_{12}$ due to the time-translational symmetry. To solve the real-time Green's functions self-consistently, we introduce the spectral function as

$$
G^R(\omega) = \int \mathrm{d}z \frac{\rho(z)}{z - \omega + i0},
\tag{3}
$$

which implies $\rho(\omega) = -\mathrm{Im}G^R(\omega)/\pi$. Other Green's functions are determined by the fluctuation-dissipation theorem as

$$
\begin{aligned}
G^{<}(\omega) &= 2\pi i n_F(\omega)\rho(\omega), \\
G^{>}(\omega) &= -2\pi i n_F(-\omega)\rho(\omega),
\end{aligned}
\tag{4}
$$

where $n_F(\omega)$ is the Fermi-Dirac distribution function. For later convenience, we introduce the Keldysh Green's function

$$
\begin{aligned}
G^K(\omega) &\equiv G^{>}(\omega) + G^{<}(\omega) \\
&= (2n_F(\omega) - 1)\, 2\pi\rho(\omega) \\
&= (1 - 2n_F(\omega))\left(G^R(\omega) - G^A(\omega)\right)
\end{aligned}
\tag{5}
$$

———————

* PengfeiZhang.physics@gmail.com

To derive the self-consistent equation for the spectral function, we the Schwinger-Dyson equation in real time reads

$$G^R(\omega)^{-1} = \omega + \frac{h}{2}\sigma^x - \Sigma^R(\omega) \tag{6}$$

, with real-time self-energies obtained via large-$N$ and the melon diagram approximations are given by

$$\begin{aligned}
\Sigma^{\gtrless}(t_1,t_2) &= \frac{J^2}{4}\sum_{\alpha,\alpha'}\xi_\alpha\xi_{\alpha'} \\
&\sigma^{\alpha'}G^{\gtrless}(t_1,t_2)\sigma^\alpha \mathrm{Tr}\left[\sigma^{\alpha'}G^{\gtrless}(t_1,t_2)\sigma^\alpha G^{\lessgtr}(t_2,t_1)\right],
\end{aligned} \tag{7}$$

Here $\Sigma^{\gtrless}, \sigma^\alpha$ and $G^{\gtrless}$ are all $2\times 2$ matrix corresponding to spin indexes, and trace operation is applied on the spin indexes. Furthermore, $\Sigma^{R/A}$ and $\Sigma^K$ related to $\Sigma^{\gtrless}$ as

$$\begin{aligned}
\Sigma^{R/A}(t_1,t_2) &= \pm\Theta\left(\pm t_{12}\right)\left(\Sigma^>(t_1,t_2) - \Sigma^<(t_1,t_2)\right), \\
\Sigma^K(t_1,t_2) &= \left(\Sigma^>(t_1,t_2) + \Sigma^<(t_1,t_2)\right).
\end{aligned} \tag{8}$$

Iteratively solving Eq. (6) gives the spectral function $\rho_{ss'}(\omega)(s,s' =\uparrow,\downarrow)$ and real-time Green's functions.

In numerics, we choose the discretization of time domain $t$ and frequency domain $\omega$ to be 3001. The cutoff in time domain is $[-\Lambda_t,\Lambda_t] = [-20.0,20.0]$, and frequency domain is time domain is $[-\Lambda_\omega,\Lambda_\omega] = [-12.0,12.0]$.

## II. THE SYMMETRY OF THE GREEN'S FUNCTION

In this section, we analyze the symmetries of the fermion Green's function encoded in random spin model

$$\hat{H}(t) = \sum_{1\leq i<j\leq N} J_{ij}(\hat{S}_i^x\hat{S}_j^x + \xi_2\hat{S}_i^y\hat{S}_j^y + \xi_3\hat{S}_i^z\hat{S}_j^z) - h(t)\sum_{1\leq i\leq N}\hat{S}_i^x. \tag{9}$$

There are two symmetries crucial for the later simplification. The first can be regarded as $\pi$ rotation of axis $x$.

$$\begin{aligned}
\hat{c}_{s_1} &\to \sum_{s'}\left(i\sigma^x\right)_{s_1 s'}\hat{c}_{s'} \\
\hat{c}_{s_1}^\dagger &\to \sum_{s'}\hat{c}_{s'}^\dagger\left(-i\sigma^x\right)_{s' s_1}.
\end{aligned} \tag{10}$$

With symmetry in Eq. (10), $\left\{\hat{S}^x, \hat{S}^y, \hat{S}^z\right\}$ is mapped to $\left\{\hat{S}^x, -\hat{S}^y, -\hat{S}^z\right\}$, and therefore keeps Eq. (9) invariant. As a consequence, the symmetry of Green's function leads to

$$\begin{aligned}
G^>_{s_1 s_2}(t_1,t_2) &= -i\langle\hat{c}_{s_1}(t_1)\hat{c}_{s_2}^\dagger(t_2)\rangle \\
&= \sum_{s' s''}\sigma^x_{s_1 s'}G^>_{s' s''}(t_1,t_2)\sigma^x_{s'' s_2} \\
&= \begin{pmatrix} G^>_{\downarrow\downarrow}(t_1,t_2) & G^>_{\downarrow\uparrow}(t_1,t_2) \\ G^>_{\uparrow\downarrow}(t_1,t_2) & G^>_{\uparrow\uparrow}(t_1,t_2) \end{pmatrix}_{s_1 s_2}
\end{aligned} \tag{11}$$

The second symmetry is combined with particle-hole symmetry and rotation, which reads as

$$\begin{aligned}
\hat{c}_{s_1} &\to \sum_{s'}\left(i\sigma^y\right)_{s_1 s'}\hat{c}_{s'}^\dagger \\
\hat{c}_{s_1}^\dagger &\to \sum_{s'}\hat{c}_{s'}\left(-i\sigma^y\right)_{s' s_1}.
\end{aligned} \tag{12}$$

Similarly, with symmetry in Eq. (12), $\left\{\hat{S}^x, \hat{S}^y, \hat{S}^z\right\}$ is mapped to $\left\{\hat{S}^x, -\hat{S}^y, \hat{S}^z\right\}$, and therefore keeps Eq. (9) invariant. Also, the symmetry of Green's function leads to

$$\begin{aligned}
G^>_{s_1 s_2}(t_1,t_2) &= -i\langle\hat{c}_{s_1}(t_1)\hat{c}_{s_2}^\dagger(t_2)\rangle \\
&= -\sum_{s' s''}\sigma^y_{s_1 s'}G^<_{s'' s'}(t_2,t_1)\sigma^y_{s'' s_2} \\
&= \begin{pmatrix} -G^<_{\uparrow\uparrow}(t_2,t_1) & G^<_{\uparrow\downarrow}(t_2,t_1) \\ G^<_{\uparrow\downarrow}(t_2,t_1) & -G^<_{\uparrow\uparrow}(t_2,t_1) \end{pmatrix}_{s_1 s_2},
\end{aligned} \tag{13}$$

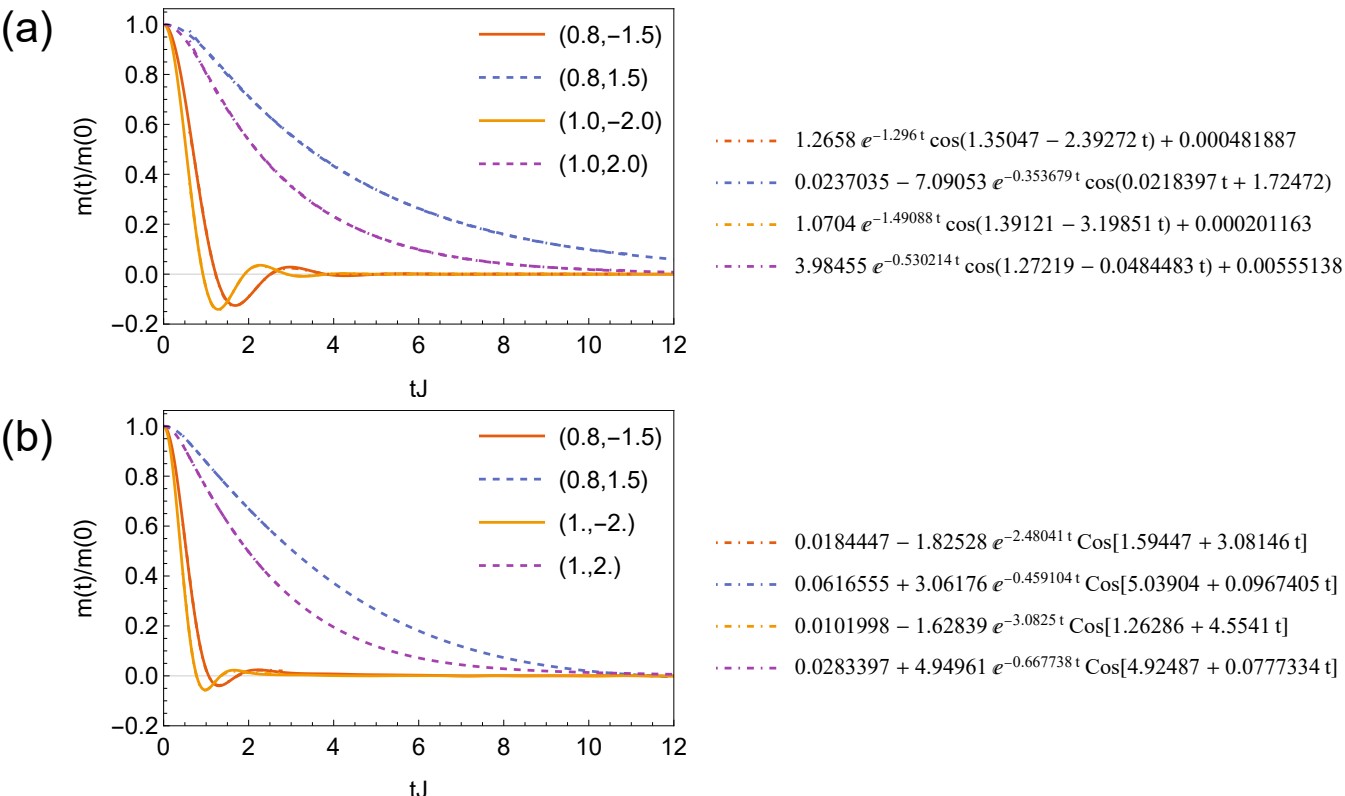

FIG. 1. Fitting of (a) the large-$N$ dynamics; (b) the exact diagonalization dynamics corresponding to the main text fig. (2).

where the second line to the third line uses the symmetry obtained in Eq. (11). Finally, we can exchange $>$ and $<$ symbols in Eq. (11) and Eq. (11) to obtain another two symmetry in terms of Green's function.

## III.  DETAILS OF THE NUMERICAL SIMULATION

### A.  Kadanoff-Baym equation

We choose the discretiation of Green's function matrix $G^{\gtrless}(t_1, t_2)$ to be $4001 \times 4001$, with a cutoff $t_1, t_2 \in [-\Lambda_t, \Lambda_t] = [-20.0, 20.0]$. As we discussed in the main text, for $t_1, t_2 < 0$, the system is prepared in thermal equilibrium with $G^{\gtrless}(t_1, t_2) = G^{\gtrless}_\beta(t_{12})$.

To obtain the oscillation frequency, we fit the magnetization dynamics with the formula

$$m(t) = m_0 \cos(\Omega t + \theta) e^{-\Gamma t} + m_{\text{offset}}. \tag{14}$$

Here $m_0$ is the amplitude, $\theta$ is the phase, $\Gamma$ is the quasi-particle decay rate, and $m_{\text{offset}}$ is the offset which is significant in the finite $N$ exact diagonalization numerics.

We observe that the offset $m_{\text{offset}}$ is negligibly small in the order of $10^{-3}$ when the fitting region covers enough long time, which is consistent with the symmetry reason. Therefore, we choose fitting region $tJ \in [0.61, 11]$ in fig. 1(a) to ensure the correctness of fitting. We have also checked that the fitting parameters are not sensitive to the detailed fitting region if it covers enough early time and long time regions. In numerics, we use Mathematica `NonlinearModelFit` function to perform the non-linear fitting.

### B.  Exact Diagonalization

Due to the finite $N$ correction, we find the result deviates from the formula Eq. (14) if fitting includes the entire long-time and early-time region. Therefore, we aim to focus on the small $m(t)/m(0)$ region where oscillations happen.

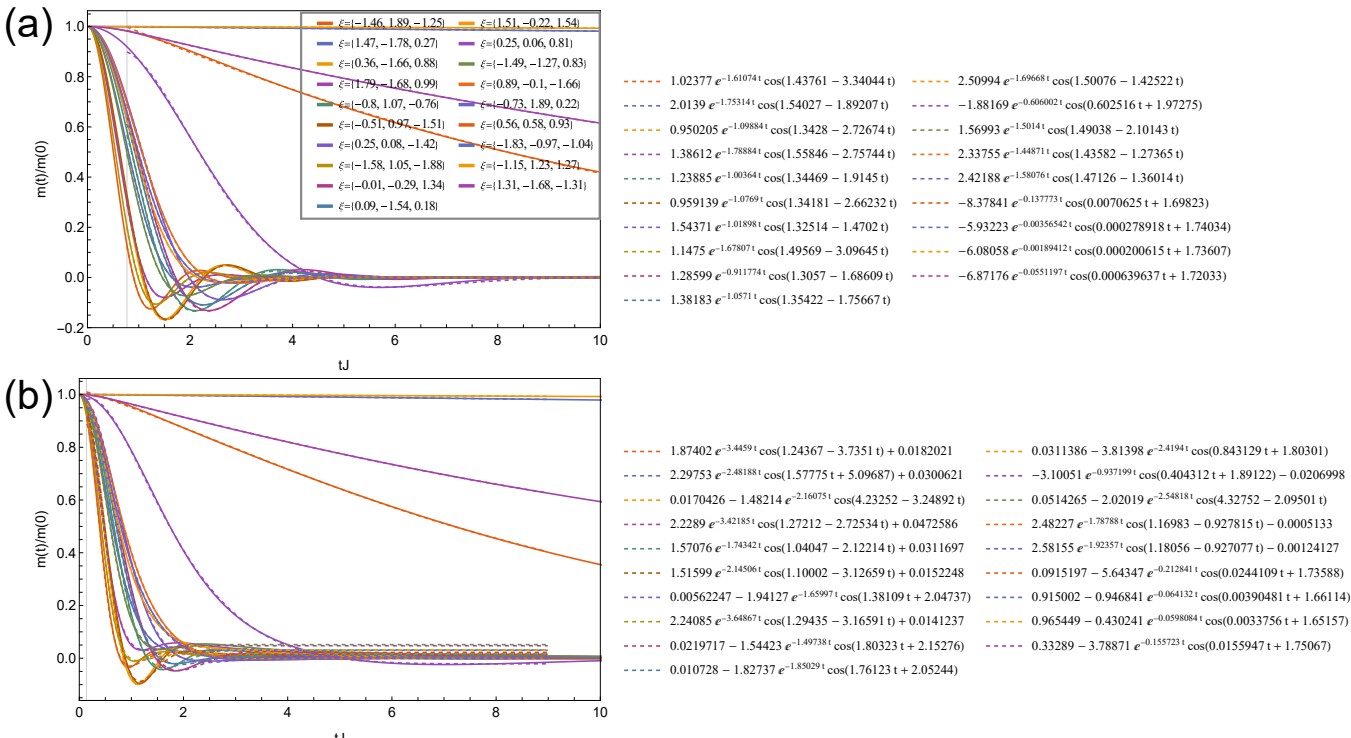

FIG. 2. Fitting of (a) the large-$N$ dynamics; (b) the exact diagonalization dynamics corresponding to the main text fig. (3). The solid line is the numerical data, and the dashed line is the fitting curve. The inset legend lists each anisotropic parameter corresponding to the 19 cases in the main text. Besides, the right panel shows the fitting functions for each case.

Here we take fitting region $tJ \in [0.45, 2.8]$. From fig. 1(b) we find the fitting precisely matches with the dynamics data. Although the oscillation frequency is more sensitive to the fitting region compared with the large-$N$ case, the behavior of oscillation frequency in terms of anisotropic parameter qualitatively remains no change.

## C. Estimation of frequency and error bar

Fig. (3) in the main text shows the estimation of frequency and error bars with random anisotropic parameters. Here we show the detailed fitting result in fig. 2, where the initial temperature and external magnetic field is $\beta J = 0.04$, $\bar{J} = 0$ and $h/J = 10$.

The choice of the fitting region is naturally uncertain. First, the theoretical prediction of the oscillation frequency acquires the assumption of small $m(t)$, but the small $m(t)$ time region is not uniquely defined. Secondly, especially for ED numerics, finite system size leads to untrustable results in the late time limit. Therefore it is reasonable only to consider the early time when fitting the ED numerics, which also introduces uncertainty of fitting region.

The error bars of the fitting frequency arise from such uncertainty. To quantifies such error, we separately consider two different numerical approaches.

- For the large-$N$ data, we choose the time region begins at different values: $t_{\text{begin}}J = 0, 0.05, 0.1, \cdots, 1.2$, and ends at the same point $t_{\text{end}}J = 10.0$. We fit these frequencies and take the standard deviation as the error bars.

- For the ED data, we choose the time region begins at different values: $t_{\text{begin}}J = 0, 0.08, 0.16, \cdots, 0.4$, and ends at also different points $t_{\text{end}}J = 3.0, 4.0, \cdots 9.0$. We take a combination of these beginning and ending points and take the standard deviation among these frequencies as the error bars.

We summarize the detailed fitting region in table. I. Notice that for the ED approach, we need a larger ending fitting time for the small frequency cases, but in other cases, we choose a smaller ending fitting time to avoid the finite $N$ effect.

| Cases | 1 | 2 | 3 | 4 | 5 | 6 | 7 | 8 | 9 | 10 | 11 | 12 | 13 | 14 | 15 | 16 | 17 | 18 | 19 |
|---|---|---|---|---|---|---|---|---|---|---|---|---|---|---|---|---|---|---|---|
| $\xi_1$ | -1.46 | 1.47 | 0.36 | 1.79 | -0.8 | -0.51 | 0.25 | -1.58 | -0.01 | 0.09 | 1.51 | 0.25 | -1.49 | 0.89 | -0.73 | 0.56 | -1.83 | -1.15 | 1.31 |
| $\xi_2$ | 1.89 | -1.78 | -1.66 | -1.68 | 1.07 | 0.97 | 0.08 | 1.05 | -0.29 | -1.54 | -0.22 | 0.06 | -1.27 | -0.1 | 1.89 | 0.58 | -0.97 | 1.23 | -1.68 |
| $\xi_3$ | -1.25 | 0.27 | 0.88 | 0.99 | -0.76 | -1.51 | -1.42 | -1.88 | 1.34 | 0.18 | 1.54 | 0.81 | 0.83 | -1.66 | 0.22 | 0.93 | -1.04 | 1.27 | -1.31 |
| $\Omega_{KB}$ | 3.34 | 1.89 | 2.73 | 2.76 | 1.91 | 2.66 | 1.47 | 3.10 | 1.69 | 1.76 | 1.43 | 0.60 | 2.10 | 1.27 | 1.36 | 0.01 | 0.00 | 0.00 | 0.00 |
| $\sigma_{\Omega_{KB}}$ | 0.19 | 0.17 | 0.13 | 0.20 | 0.12 | 0.13 | 0.10 | 0.20 | 0.10 | 0.12 | 0.12 | 0.03 | 0.16 | 0.10 | 0.11 | 0.01 | 0.00 | 0.00 | 0.01 |
| $\Omega_{ED}$ | 3.74 | 1.58 | 3.25 | 2.73 | 2.12 | 3.13 | 1.38 | 3.17 | 1.80 | 1.76 | 0.84 | 0.40 | 2.10 | 0.93 | 0.93 | 0.02 | 0.00 | 0.00 | 0.02 |
| $\sigma_{\Omega_{ED}}$ | 0.59 | 0.46 | 0.31 | 0.52 | 0.17 | 0.30 | 0.05 | 0.49 | 0.11 | 0.10 | 0.33 | 0.16 | 0.23 | 0.08 | 0.16 | 0.05 | 0.04 | 0.04 | 0.04 |
| $\Omega_{Th}$ | 3.53 | 1.73 | 3.04 | 2.69 | 2.08 | 2.97 | 1.55 | 3.17 | 1.85 | 1.87 | 1.22 | 0.63 | 2.07 | 1.14 | 1.19 | 0.00 | 0.00 | 0.00 | 0.00 |
| $t_{\text{begin,KB}}J$ | 0.75 | 0.75 | 0.75 | 0.75 | 0.75 | 0.75 | 0.75 | 0.75 | 0.75 | 0.75 | 0.75 | 0.75 | 0.75 | 0.75 | 0.75 | 0.75 | 0.75 | 0.75 | 0.75 |
| $t_{\text{end,KB}}J$ | 10 | 10 | 10 | 10 | 10 | 10 | 10 | 10 | 10 | 10 | 10 | 10 | 10 | 10 | 10 | 10 | 10 | 10 | 10 |
| $t_{\text{begin,ED}}J$ | 0.15 | 0.15 | 0.15 | 0.15 | 0.15 | 0.15 | 0.15 | 0.15 | 0.15 | 0.15 | 0.15 | 0.15 | 0.15 | 0.15 | 0.15 | 0.15 | 0.15 | 0.15 | 0.15 |
| $t_{\text{end,ED}}J$ | 3.5 | 3.5 | 3.5 | 3.5 | 3.5 | 3.5 | 3.5 | 3.5 | 3.5 | 3.5 | 3.5 | 9 | 3.5 | 9 | 9 | 9 | 9 | 9 | 9 |

TABLE I. The detailed parameters and result for each case. $(\xi_1, \xi_2, \xi_3)$ means anisotropic parameters. $\Omega_{KB}, \Omega_{ED}, \Omega_{Th}$ are the fitting frequencies from Kadanoff-Baym numerics, ED numerics, and theoretical prediction. $\sigma_{\Omega_{KB}}, \sigma_{\Omega_{ED}}$ are the standard deiviations for KB and ED numerics. $[t_{\text{begin,KB}}J, t_{\text{end,KB}}J]$ and $[t_{\text{begin,ED}}J, t_{\text{end,ED}}J]$ are the fit region which lead to the $\Omega_{KB}, \Omega_{ED}$ numerics data of the main text fig. 3.

## IV. EXPLICIT CALCULATION OF THE SELF-ENERGY

Starting from Eq. (7), we derivate the corresponding analytical formula. We first consider the trace part $\text{Tr}\left[\sigma^{\alpha'}G^{\gtrless}(t_1,t_2)\sigma^{\alpha}G^{\lessgtr}(t_2,t_1)\right]$. In the basis of $\alpha', \alpha = \{x,y,z\}$, trace part becomes

$$
\begin{aligned}
&\text{Tr}\left[\sigma^{\alpha'}G^{\gtrless}(t_1,t_2)\sigma^{\alpha}G^{\lessgtr}(t_2,t_1)\right] \\
&= -\text{Tr}\left[\sigma^y\sigma^{\alpha'}G^{\gtrless}(t_1,t_2)\sigma^{\alpha}\sigma^y G^{\gtrless}(t_1,t_2)\right] \\
&= \begin{pmatrix}
-2G^{\gtrless}_{\uparrow\uparrow}(t_1,t_2)^2 + 2G^{\gtrless}_{\uparrow\downarrow}(t_1,t_2)^2 & 0 & 0 \\
0 & -2G^{\gtrless}_{\uparrow\uparrow}(t_1,t_2)^2 - 2G^{\gtrless}_{\uparrow\downarrow}(t_1,t_2)^2 & 4iG^{\gtrless}_{\uparrow\uparrow}(t_1,t_2)G^{\gtrless}_{\uparrow\downarrow}(t_1,t_2) \\
0 & -4iG^{\gtrless}_{\uparrow\uparrow}(t_1,t_2)G^{\gtrless}_{\uparrow\downarrow}(t_1,t_2) & -2G^{\gtrless}_{\uparrow\uparrow}(t_1,t_2)^2 - 2G^{\gtrless}_{\uparrow\downarrow}(t_1,t_2)^2
\end{pmatrix}_{\alpha'\alpha}.
\end{aligned}
\tag{15}
$$

The second line applies the symmetry Eq. (13) and implicitly use Eq. (11) to ensure $\left(G^{\gtrless}\right)^T = G^{\gtrless}$. The disappearance of off-diagonal certain matrix elements is non-trivial. For example, we take $\alpha' = x, \alpha = z$.

$$
\begin{aligned}
&-\text{Tr}\left[\sigma^y\sigma^x G^{\gtrless}(t_1,t_2)\sigma^z\sigma^y G^{\gtrless}(t_1,t_2)\right] \\
&= -\text{Tr}\left[\sigma^y G^{\gtrless}(t_1,t_2)\sigma^x\sigma^z\sigma^y G^{\gtrless}(t_1,t_2)\right] \\
&= i\text{Tr}\left[\sigma^y G^{\gtrless}(t_1,t_2)G^{\gtrless}(t_1,t_2)\right] \\
&= i\text{Tr}\left[\sigma^y \begin{pmatrix} G^{\gtrless}_{\uparrow\uparrow}(t_1,t_2)^2 + G^{\gtrless}_{\uparrow\downarrow}(t_1,t_2)^2 & 2G^{\gtrless}_{\uparrow\uparrow}(t_1,t_2)G^{\gtrless}_{\uparrow\downarrow}(t_1,t_2) \\ 2G^{\gtrless}_{\uparrow\uparrow}(t_1,t_2)G^{\gtrless}_{\uparrow\downarrow}(t_1,t_2) & G^{\gtrless}_{\uparrow\uparrow}(t_1,t_2)^2 + G^{\gtrless}_{\uparrow\downarrow}(t_1,t_2)^2 \end{pmatrix}\right].
\end{aligned}
$$

Since the non-zero elements in Pauli matrix $\sigma^y$ are $\pm i$ seperately, we find $\text{Tr}\left[\sigma^y G^{\gtrless}(t_1,t_2)G^{\gtrless}(t_1,t_2)\right] = 0$. Similarly, the $\alpha' = x, \alpha = y$ case corresponds to $\text{Tr}\left[\sigma^z G^{\gtrless}(t_1,t_2)G^{\gtrless}(t_1,t_2)\right] = 0$ and therefore the trace part vanishes.

However, it is special for $\alpha' = y, \alpha = z$, which reduced to the final result $\text{Tr}\left[\sigma^x G^{\gtrless}(t_1,t_2)G^{\gtrless}(t_1,t_2)\right] \neq 0$, since $\sigma^x$ only contains two 1 elements. Hence, the only non-zero off-diagonal trace part is $\alpha' = y, \alpha = z$, and $\alpha' = z, \alpha = y$.

To further calculate the self-energy, we show each non-zero contribution in Eq. (7).

a. $\alpha' = x, \alpha = x$

$$
\frac{1}{2}J^2\xi_1^2 \begin{pmatrix} G^{\gtrless}_{\uparrow\uparrow}(t_1,t_2) & G^{\gtrless}_{\uparrow\downarrow}(t_1,t_2) \\ G^{\gtrless}_{\uparrow\downarrow}(t_1,t_2) & G^{\gtrless}_{\uparrow\uparrow}(t_1,t_2) \end{pmatrix} \left(-G^{\gtrless}_{\uparrow\uparrow}(t_1,t_2)^2 + G^{\gtrless}_{\uparrow\downarrow}(t_1,t_2)^2\right)
\tag{16}
$$

b. $\alpha' = y, \alpha = y$

$$
\frac{1}{2}J^2\xi_2^2 \begin{pmatrix} -G^{\gtrless}_{\uparrow\uparrow}(t_1,t_2) & G^{\gtrless}_{\uparrow\downarrow}(t_1,t_2) \\ G^{\gtrless}_{\uparrow\downarrow}(t_1,t_2) & -G^{\gtrless}_{\uparrow\uparrow}(t_1,t_2) \end{pmatrix} \left(G^{\gtrless}_{\uparrow\uparrow}(t_1,t_2)^2 + G^{\gtrless}_{\uparrow\downarrow}(t_1,t_2)^2\right)
\tag{17}
$$

c. $\alpha' = z, \alpha = z$

$$\frac{1}{2}J^2\xi_3^2 \begin{pmatrix} -G_{\uparrow\uparrow}^{\gtrless}(t_1,t_2) & G_{\uparrow\downarrow}^{\gtrless}(t_1,t_2) \\ G_{\uparrow\downarrow}^{\gtrless}(t_1,t_2) & -G_{\uparrow\uparrow}^{\gtrless}(t_1,t_2) \end{pmatrix} \left(G_{\uparrow\uparrow}^{\gtrless}(t_1,t_2)^2 + G_{\uparrow\downarrow}^{\gtrless}(t_1,t_2)^2\right) \tag{18}$$

d. $\alpha' = y, \alpha = z$

$$J^2\xi_2\xi_3 \begin{pmatrix} G_{\uparrow\downarrow}^{\gtrless}(t_1,t_2) & -G_{\uparrow\uparrow}^{\gtrless}(t_1,t_2) \\ -G_{\uparrow\uparrow}^{\gtrless}(t_1,t_2) & G_{\uparrow\downarrow}^{\gtrless}(t_1,t_2) \end{pmatrix} \left(G_{\uparrow\uparrow}^{\gtrless}(t_1,t_2)G_{\uparrow\downarrow}^{\gtrless}(t_1,t_2)\right) \tag{19}$$

e. $\alpha' = z, \alpha = y$

$$J^2\xi_2\xi_3 \begin{pmatrix} G_{\uparrow\downarrow}^{\gtrless}(t_1,t_2) & -G_{\uparrow\uparrow}^{\gtrless}(t_1,t_2) \\ -G_{\uparrow\uparrow}^{\gtrless}(t_1,t_2) & G_{\uparrow\downarrow}^{\gtrless}(t_1,t_2) \end{pmatrix} \left(G_{\uparrow\uparrow}^{\gtrless}(t_1,t_2)G_{\uparrow\downarrow}^{\gtrless}(t_1,t_2)\right) \tag{20}$$

Finally, we arrive the full self-energy

$$\Sigma^{\gtrless}(t_1,t_2) = \frac{1}{2}J^2 \begin{pmatrix} -\boldsymbol{\xi}^2 G_{\uparrow\uparrow}^{\gtrless}(t_1,t_2)^3 + AG_{\uparrow\uparrow}^{\gtrless}(t_1,t_2)G_{\uparrow\downarrow}^{\gtrless}(t_1,t_2)^2 & AG_{\uparrow\uparrow}^{\gtrless}(t_1,t_2)^2 G_{\uparrow\downarrow}^{\gtrless}(t_1,t_2) + \boldsymbol{\xi}^2 G_{\uparrow\downarrow}^{\gtrless}(t_1,t_2)^3 \\ AG_{\uparrow\uparrow}^{\gtrless}(t_1,t_2)^2 G_{\uparrow\downarrow}^{\gtrless}(t_1,t_2) + \boldsymbol{\xi}^2 G_{\uparrow\downarrow}^{\gtrless}(t_1,t_2)^3 & -\boldsymbol{\xi}^2 G_{\uparrow\uparrow}^{\gtrless}(t_1,t_2)^3 + AG_{\uparrow\uparrow}^{\gtrless}(t_1,t_2)G_{\uparrow\downarrow}^{\gtrless}(t_1,t_2)^2 \end{pmatrix}, \tag{21}$$

where the polynomials $\boldsymbol{\xi}^2 = \xi_1^2 + \xi_2^2 + \xi_3^2$ and $A = -\xi_1^2 + \xi_2^2 - 4\xi_2\xi_3 + \xi_3^2$.

## V.  DERIVATION OF THE NORMAL MODE EQUATION WITH REGULAR INTERACTIONS

### A.  Derivation of Eq. (6) in the main text

As formulated in the main text, the perturbation is applied on $(G^a)_{ss'} = \left(G^{a,\beta_f}\right)_{ss'} + \delta G_{\uparrow\downarrow}^a \left(\delta_{s\uparrow}\delta_{s'\downarrow} + \delta_{s'\uparrow}\delta_{s\downarrow}\right)$, with causality index $a = \gtrless$, and the perturbation assumption $\delta G_{\uparrow\downarrow}^a \ll G_{ss}^a$ where $s = \uparrow, \downarrow$.

According to the Lorentzian approximation

$$G_{\uparrow\uparrow}^{R/A,\beta_f}(t) \approx \mp i\Theta(\pm t)e^{-\Gamma|t|/2}, \qquad G_{\uparrow\uparrow}^{\gtrless,\beta_f}(t) \approx \mp ie^{-\Gamma|t|/2}/2. \tag{22}$$

We have $G^{K,\beta_f} = G^{>,\beta_f} + G^{<,\beta_f} \approx 0$. Furthermore, we can assume $\delta G_{\uparrow\downarrow}^{\gtrless} = \frac{1}{2}\delta G_{\uparrow\downarrow}^K$, which is consistent with the definition of Keldysh Green's function in Eq. (5).

We consider the self-consistent condition

$$G^K = G^R \circ \Sigma^K \circ G^A. \tag{23}$$

The perturbation in the off-diagonal element reads as

$$G_{\uparrow\downarrow}^K = \sum_{ss'=\uparrow,\downarrow} G_{\uparrow s}^R \circ \Sigma_{ss'}^K \circ G_{s'\downarrow}^A. \tag{24}$$

We discuss the result separately

a. $s = \uparrow, s' = \uparrow$

$$\begin{aligned} \Sigma_{\uparrow\uparrow}^K &= \Sigma_{\uparrow\uparrow}^> + \Sigma_{\uparrow\uparrow}^< \\ &= \frac{1}{2}J^2 A \left(G_{\uparrow\uparrow}^>(t_1,t_2)G_{\uparrow\downarrow}^>(t_1,t_2)^2 + G_{\uparrow\uparrow}^<(t_1,t_2)G_{\uparrow\downarrow}^<(t_1,t_2)^2\right) - \frac{1}{2}J^2\boldsymbol{\xi}^2 \left(G_{\uparrow\uparrow}^>(t_1,t_2)^3 + G_{\uparrow\uparrow}^<(t_1,t_2)^3\right) \\ &= 0\left(\delta G_{\uparrow\downarrow}^{\gtrless}\right)^2 + 0 \end{aligned} \tag{25}$$

The 0 is due to the form of high temperature Green's function Eq. (22), and $G_{\uparrow\downarrow}^A$ in the Eq. (24) is the first order perturbation. Therefore this term vanishes.

*b.* $s =\uparrow, s' =\downarrow$

$$
\begin{aligned}
\Sigma_{\uparrow\downarrow}^K &= \Sigma_{\uparrow\downarrow}^> + \Sigma_{\uparrow\downarrow}^< \\
&= \frac{1}{2}J^2 A \left( G_{\uparrow\uparrow}^>(t_1,t_2)^2 G_{\uparrow\downarrow}^>(t_1,t_2) + G_{\uparrow\uparrow}^<(t_1,t_2)^2 G_{\uparrow\downarrow}^<(t_1,t_2) \right) + \frac{1}{2}J^2\boldsymbol{\xi}^2 \left( G_{\uparrow\downarrow}^>(t_1,t_2)^3 + G_{\uparrow\downarrow}^<(t_1,t_2)^3 \right) \\
&= \frac{1}{2}J^2 A \left( G_{\uparrow\uparrow}^{>,\beta_f}(t_1,t_2)^2 \delta G_{\uparrow\downarrow}^>(t_1,t_2) + G_{\uparrow\uparrow}^{<,\beta_f}(t_1,t_2)^2 \delta G_{\uparrow\downarrow}^<(t_1,t_2) \right) + \frac{1}{2}J^2\boldsymbol{\xi}^2 \left( \left(\delta G_{\uparrow\downarrow}^>(t_1,t_2)\right)^3 + \left(\delta G_{\uparrow\downarrow}^<(t_1,t_2)\right)^3 \right) \\
&\approx \frac{1}{2}J^2 A \left( G_{\uparrow\uparrow}^{>,\beta_f}(t_1,t_2)^2 \delta G_{\uparrow\downarrow}^>(t_1,t_2) + G_{\uparrow\uparrow}^{<,\beta_f}(t_1,t_2)^2 \delta G_{\uparrow\downarrow}^<(t_1,t_2) \right) \\
&= \frac{1}{4}J^2 A \left( G_{\uparrow\uparrow}^{>,\beta_f}(t_1,t_2)^2 + G_{\uparrow\uparrow}^{<,\beta_f}(t_1,t_2)^2 \right) \delta G_{\uparrow\downarrow}^K(t_1,t_2)
\end{aligned}
\tag{26}
$$

The approximate equal sign means to keep the first order correction of $\delta G_{\uparrow\downarrow}^a$ and ignore the high order term. Then we find the zero's order of $\Sigma_{\uparrow\downarrow}^K$ is zero, and therefore $\delta\Sigma_{\uparrow\downarrow}^K = \frac{1}{4}J^2 A \left( G_{\uparrow\uparrow}^{>,\beta_f}(t_1,t_2)^2 + G_{\uparrow\uparrow}^{<,\beta_f}(t_1,t_2)^2 \right) \delta G_{\uparrow\downarrow}^K(t_1,t_2)$ leads to the same result in the main text Eq. (6). Also, we notice that in this case $G_{\uparrow s}^R, G_{s'\downarrow}^A$ are both zero's order, which satisfies the linearized equation $\delta G_{\uparrow\downarrow}^K = G_{\uparrow\uparrow}^{R,\beta_f} \circ \delta\Sigma_{\uparrow\downarrow}^K \circ G_{\uparrow\uparrow}^{A,\beta_f}$.

*c.* $s =\downarrow, s' =\uparrow$    Similar to the case *(b)*.

*d.* $s =\downarrow, s' =\downarrow$    Similar to the case *(a)*.

## B. Normal modes

Starting from Eq. (6) in the main text, we apply the Lorentz approximation

$$
\begin{aligned}
G_{ss'}^{\lessgtr}(t) &= \pm\frac{i}{2}e^{-\frac{|t|\Gamma}{2}}\delta_{ss'} \\
G_{ss'}^{R/A}(t) &= \pm i e^{\mp\frac{t\Gamma}{2}}\Theta(\pm t)\delta_{ss'} \\
G_{ss'}^K(t) &= G_{ss'}^>(t) + G_{ss'}^<(t) = 0
\end{aligned}
\tag{27}
$$

It leads to the convolution equation in the time domain

$$
\begin{aligned}
e^{-\frac{\lambda}{2}(t_1+t_4)}\varphi(t_{14}) &= G_{\uparrow\uparrow}^R \circ \left( -\frac{1}{8}J^2 A e^{-|t_2-t_3|\Gamma - \frac{\lambda}{2}(t_2+t_3)}\varphi(t_{23}) \right) \circ G_{\downarrow\downarrow}^A \\
&= -\frac{1}{8}J^2 A \int dt_2\, dt_3\, e^{-\frac{\Gamma}{2}(t_1-t_2)}\Theta(t_1-t_2)e^{-|t_2-t_3|\Gamma - \frac{\lambda}{2}(t_2+t_3)}e^{\frac{\Gamma}{2}(t_3-t_4)}\Theta(-t_3+t_4)\varphi(t_{23}).
\end{aligned}
\tag{28}
$$

Here $A = -\xi_1^2 + \xi_2^2 - 4\xi_2\xi_3 + \xi_3^2$ is the polynomial in terms of the anisotropic parameters. Rearrange the variable

$$
e^{-\frac{\lambda-\Gamma}{2}(t_1+t_4)}\varphi(t_{14}) = -\frac{1}{8}J^2 A \int dt_2\, dt_3\, \Theta(t_1-t_2)\Theta(-t_3+t_4)e^{-|t_2-t_3|\Gamma - \frac{\lambda-\Gamma}{2}(t_2+t_3)}\varphi(t_{23})
\tag{29}
$$

Apply $\partial_{t_1}$

$$
e^{-\frac{\lambda-\Gamma}{2}(t_1+t_4)}\left[ -\frac{\lambda-\Gamma}{2}\varphi(t_{14}) + \partial_{t_1}\varphi(t_{14}) \right] = -\frac{1}{8}J^2 A \int dt_2\, dt_3\, \delta(t_1-t_2)\Theta(-t_3+t_4)e^{-|t_2-t_3|\Gamma - \frac{\lambda-\Gamma}{2}(t_2+t_3)}\varphi(t_{23})
\tag{30}
$$

Apply $\partial_{t_4}$

$$
\begin{aligned}
e^{-\frac{\lambda-\Gamma}{2}(t_1+t_4)}\left[ \left(\frac{\lambda-\Gamma}{2}\right)^2 \varphi(t_{14}) + \partial_{t_4}\partial_{t_1}\varphi(t_{14}) \right] &= -\frac{1}{8}J^2 A \int dt_2\, dt_3\, \delta(t_1-t_2)\delta(-t_3+t_4)e^{-|t_2-t_3|\Gamma - \frac{\lambda-\Gamma}{2}(t_2+t_3)}\varphi(t_{23}) \\
&= -\frac{1}{8}J^2 A e^{-|t_1-t_4|\Gamma - \frac{\lambda-\Gamma}{2}(t_1+t_4)}\varphi(t_{14})
\end{aligned}
\tag{31}
$$

Therefore we finally arrive at the normal mode equations

$$
-\frac{(\Gamma-\lambda)^2}{4}\varphi(t_{14}) = -\partial_{t_{14}}^2\varphi(t_{14}) + \frac{A}{8}J^2 e^{-\Gamma|t_{14}|}\varphi(t_{14}),
\tag{32}
$$

As discussed in the main text, the positive and negative of $A$ determine the oscillation and non-oscillation behaviors in the quench dynamics. For clarity, the contour plot of $A$ is shown in fig. 3.

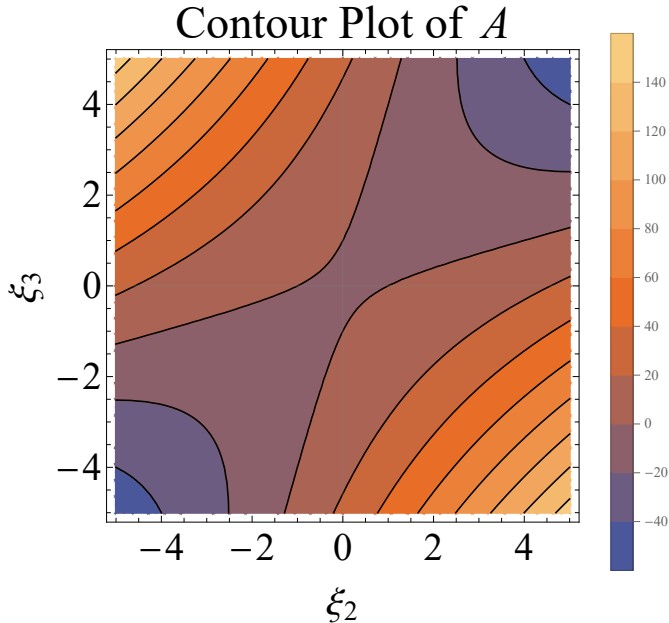

FIG. 3. The contour plot of polynomial $A$. The legend shows the $A > 0$ and $A < 0$ region.

## VI.  DERIVATION OF THE NORMAL MODE EQUATION WITH BROWNIAN INTERACTIONS

To check the effect of coherence, we place $J_{ij}$ in the Hamiltonian Eq. (9) with Brownian variables $J_{ij}(t)$, where $\overline{J_{ij}(t_1)J_{ij}(t_2)} = 4J^2/N\delta(t_1 - t_2)$. The self energy is replaced by

$$\Sigma^{\gtrless}(t_1,t_2) = \frac{1}{2}J^2\delta(t_1-t_2)\begin{pmatrix} -\boldsymbol{\xi}^2 G_{\uparrow\uparrow}^{\gtrless}(t_1,t_2)^3 + AG_{\uparrow\uparrow}^{\gtrless}(t_1,t_2)G_{\uparrow\downarrow}^{\gtrless}(t_1,t_2)^2 & AG_{\uparrow\uparrow}^{\gtrless}(t_1,t_2)^2 G_{\uparrow\downarrow}^{\gtrless}(t_1,t_2) + \boldsymbol{\xi}^2 G_{\uparrow\downarrow}^{\gtrless}(t_1,t_2)^3 \\ AG_{\uparrow\uparrow}^{\gtrless}(t_1,t_2)^2 G_{\uparrow\downarrow}^{\gtrless}(t_1,t_2) + \boldsymbol{\xi}^2 G_{\uparrow\downarrow}^{\gtrless}(t_1,t_2)^3 & -\boldsymbol{\xi}^2 G_{\uparrow\uparrow}^{\gtrless}(t_1,t_2)^3 + AG_{\uparrow\uparrow}^{\gtrless}(t_1,t_2)G_{\uparrow\downarrow}^{\gtrless}(t_1,t_2)^2 \end{pmatrix}, \tag{33}$$

Therefore, following the same steps, the linearization of the Schwinger-Dyson equation $G^K = G^R \circ \Sigma^K \circ G^A$ leads to

$$\delta G_{\uparrow\downarrow}^K(t_1,t_1) = \left[ G_{\uparrow\uparrow}^{R,\beta_f} \circ \delta\Sigma_{\uparrow\downarrow}^K \circ G_{\uparrow\uparrow}^{A,\beta_f} \right](t_1,t_1),$$

$$\delta\Sigma_{\uparrow\downarrow}^K(t_1,t_2) = \frac{1}{4}J^2 A\delta(t_1-t_2)\left( G_{\uparrow\uparrow}^{>,\beta_f}(t_1,t_1)^2 + G_{\uparrow\uparrow}^{<,\beta_f}(t_1,t_1)^2 \right)\delta G_{\uparrow\downarrow}^K(t_1,t_1). \tag{34}$$

To form close equation groups, we set the time argument of the first equation to be $(t_1, t_1)$, and then we study the equal time perturbation of the Green's function $\delta G_{\uparrow\downarrow}^K(t,t)$.

Using the high-temperature solution, we get

$$\delta G_{\uparrow\downarrow}(t_1,t_1) = \int dt_2 \; G_{\uparrow\uparrow}^R(t_1,t_2)\left( -\frac{1}{8}J^2 A e^{-|t_2-t_2|\Gamma}\delta G_{\uparrow\downarrow}(t_2,t_2) \right) G_{\downarrow\downarrow}^A(t_2,t_1)$$

$$= -\frac{1}{8}J^2 A \int dt_2 e^{-\frac{\Gamma}{2}(t_1-t_2)}\Theta(t_1-t_2)\delta G_{\uparrow\downarrow}(t_2,t_2)e^{\frac{\Gamma}{2}(t_2-t_1)}\Theta(-t_2+t_1). \tag{35}$$

$$= -\frac{1}{8}J^2 A \int dt_2 e^{-\Gamma(t_1-t_2)}\Theta(t_1-t_2)\delta G_{\uparrow\downarrow}(t_2,t_2).$$

Multiply $e^{\Gamma t_1}$ and take $\partial_{t_1}$ on the Eq. (35), which gives

$$\partial_{t_1}\left[ e^{\Gamma t_1}\delta G_{\uparrow\downarrow}(t_1,t_1) \right] = -\frac{1}{8}J^2 A \int dt_2 \; e^{\Gamma t_2}\delta(t_1-t_2)\delta G_{\uparrow\downarrow}(t_2,t_2). \tag{36}$$

Finally leads to

$$\left( \partial_t + \Gamma \right)\delta G_{\uparrow\downarrow}^K(t,t) = -\frac{AJ}{8}\delta G_{\uparrow\downarrow}^K(t,t). \tag{37}$$

Solving this differential equation leads to $\delta G_{\uparrow\downarrow}^K(t,t) \sim m(t) \sim e^{-(\Gamma + AJ/8)t}$ with a simple exponential decay.

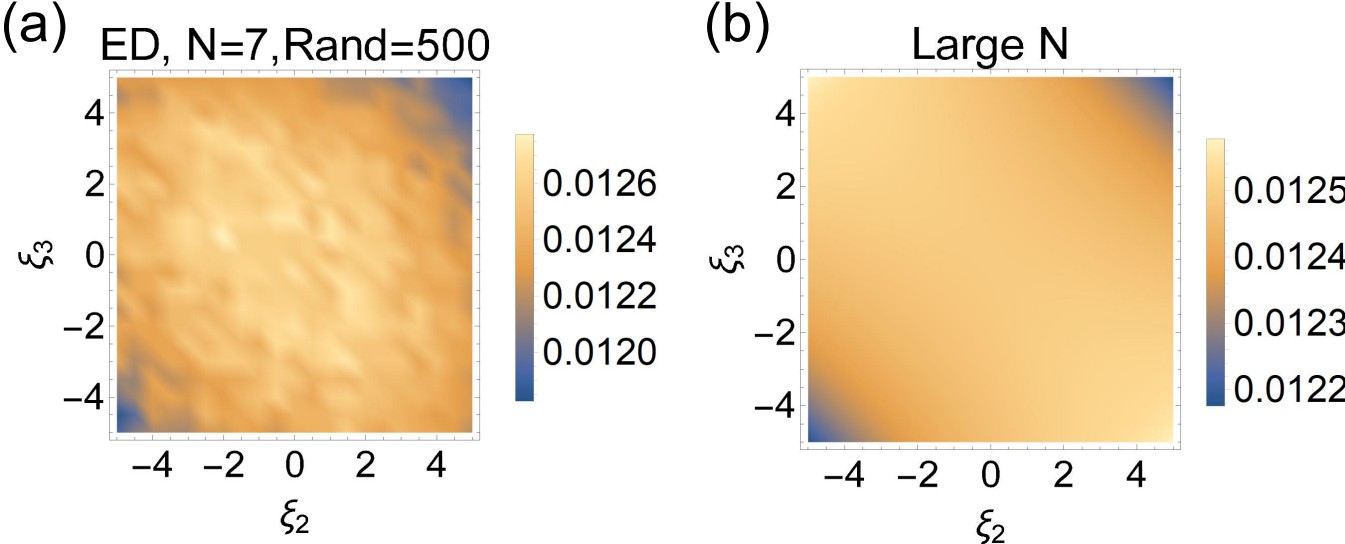

FIG. 4. (a) The susceptibility of the exact diagonalization numerics. We choose site number $N = 7$, and random realizations to be 500. (b) The susceptibility of the large-$N$ numerics. In both cases, to obtain the susceptibility, we choose the equilibrium thermal state with temperature $T/J = 10.0$, and external magnetic field with $h/J = 0.05$ in $x$ direction. We take the region of anisotropic parameter $\xi_2, \xi_3$ to be $[-5, 5]$, and the discretization step is $\Delta\xi = 0.5$.

## VII.   SUSCEPTIBILITY IN THERMAL EQUILIBRIUM

As we discussed in the main text, we calculate the equilibrium susceptibility. We take a small external magnetic field in the $x$ direction in the equilibrium thermal state. The finite difference susceptibility is defined as $\chi = \langle \hat{S}_x \rangle_h / h$, where $\langle \hat{S}_x \rangle_h$ means the thermal average in the external magnetic field $h$. First, we find the exact diagonalization and large-$N$ Kadanoff-Baym results in fig. 4 agree well with each other. Second, equilibrium susceptibility in fig. 4 is highly in contrast with the criterion for the relaxation dynamics $A$ in fig. 3. There is no appreciable distinction between $A > 0$ and $A < 0$ region correspondingly in the plot of the equilibrium. Hence, it reveals the significance of our dynamical framework.