# Peer review of "Universal Aspects of High-Temperature Relaxation Dynamics in Random Spin Models"

_SciPost Physics_

## Round 2 · Referee Report · Subir Sachdev (Referee 1) · 2024-9-18

Report

This is a well written paper containing important new results on the non-equilibrium dynamics of a strongly interacting quantum system at high temperature. The analytic analysis generalizes the Sachdev-Ye melon diagram equations to include spin anisotropy and Keldysh contours. These results are nicely compared to numerics.

I gladly recommend publication after the following change:
The discussion in the abstract and the introduction gives the reader the impression that the authors have developed a theory which is independent of the melon-diagram Kadanoff-Byam equations. Thus e.g. the abstract states:
"To validate our theory, we compare it to numerical simulations by solving the Kadanoff-Baym (KB) equation with a melon diagram approximation...."
In fact, the theory is also obtained from the KB equations with a melon diagram approximation, and this is not the impression one gets from such a sentence. It seems to me that only the ED study is beyond such an approximation.

There is a similar issue with the introduction.

I think this can be easily cleared up with changes in the language.

Recommendation

Ask for minor revision

---

## Round 2 · Referee Report · Anonymous (Referee 2) · 2024-10-10

Report

The authors calculate the relaxation dynamics of the magnetization for an infinite range anisotropic Heisenberg model staring from a high temperature initial condition. These are computed from Kadanoff Baym equations on the Schwinger Keldysh contour. These equations are linearized and numerically solved and compared to exact diagonalization. Hence it is established that the magnetization exhibits either monotonic decay or long lived oscillations, depending on a single anisotropy parameter

The results seem valid, and the investigation seems thorough. At the same time the scope seems relatively narrow. I think this can be published more or less as is, but I would think this is more suited to SciPost Physics Core.

Recommendation

Accept in alternative Journal (see Report)

---

## Editorial Decision

resubmitted